# The effects of psychosocial stress on dopaminergic function and the acute stress response

Michael AP Bloomfield[1,2,3,4,5,6,7]*, Robert A McCutcheon[1,3], Matthew Kempton[3], Tom P Freeman[2,4,8], Oliver Howes[1,3]*

[1]Psychiatric Imaging Group, MRC London Institute of Medical Sciences, Imperial College London, London, United Kingdom; [2]Translational Psychiatry Research Group, Research Department of Mental Health Neuroscience, Division of Psychiatry, UCL Institute of Mental Health, University College London, London, United Kingdom; [3]Department of Psychosis Studies, Institute of Psychiatry, Psychology and Neuroscience, Kings College London, London, United Kingdom; [4]Clinical Psychopharmacology Unit, Research Department of Clinical, Educational and Health Psychology, University College London, London, United Kingdom; [5]NIHR University College London Hospitals Biomedical Research Centre, London, United Kingdom; [6]The Traumatic Stress Clinic, St Pancras Hospital, Camden and Islington NHS Foundation Trust, London, United Kingdom; [7]National Hospital for Neurology and Neurosurgery, University College London Hospitals NHS Foundation Trust, London, United Kingdom; [8]Department of Psychology, University of Bath, Bath, United Kingdom

*For correspondence:
m.bloomfield@ucl.ac.uk (MAPB);
oliver.howes@kcl.ac.uk (OH)

**Abstract** Chronic psychosocial adversity induces vulnerability to mental illnesses. Animal studies demonstrate that this may be mediated by dopaminergic dysfunction. We therefore investigated whether long-term exposure to psychosocial adversity was associated with dopamine dysfunction and its relationship to psychological and physiological responses to acute stress. Using 3,4-dihydroxy-6-[$^{18}$F]-fluoro-$l$-phenylalanine ([$^{18}$F]-DOPA) positron emission tomography (PET), we compared dopamine synthesis capacity in $n$ = 17 human participants with high cumulative exposure to psychosocial adversity with $n$ = 17 age- and sex-matched participants with low cumulative exposure. The PET scan took place 2 hr after the induction of acute psychosocial stress using the Montréal Imaging Stress Task to induce acute psychosocial stress. We found that dopamine synthesis correlated with subjective threat and physiological response to acute psychosocial stress in the low exposure group. Long-term exposure to psychosocial adversity was associated with dampened striatal dopaminergic function (p=0.03, $d$ = 0.80) and that psychosocial adversity blunted physiological yet potentiated subjective responses to acute psychosocial stress. Future studies should investigate the roles of these changes in vulnerability to mental illnesses.
DOI: https://doi.org/10.7554/eLife.46797.001

## Introduction

Chronic psychosocial adversity increases the risk of mental illnesses including schizophrenia and depression (*van Os et al., 2010*; *Howes and Murray, 2014*; *Parker, 1983*). These adverse factors include developmental psychological trauma (*Bendall et al., 2008*) and adult life events (situations or occurrences that bring about a negative change in personal circumstances and involve threat) (*Beards et al., 2013*; *Brown and Birley, 1968*). Several lines of evidence indicate a potential

causative component to these relationships as these risk exposures demonstrate dose-response relationships (*Janssen et al., 2004*; *Morgan and Fisher, 2007*; *Pedersen and Mortensen, 2001*). Reverse causality in the form of recall bias does not appear to be driving these associations (*Cutajar et al., 2010*), and the cessation of stressor reduces the risk of illness (*Kelleher et al., 2013*). However, we lack a precise mechanistic understanding of how exposure to these risk factors induces vulnerability to mental illness, and why these exposures all increase risk. Understanding this is important to identify targets for prevention and novel treatment. One common component underlying these factors is exposure to psychosocial stress (*Howes and Murray, 2014*) via activation of the hypothalamic-pituitary-adrenal (HPA) axis and the sympathetic nervous system as part of the normal biological stress response (*Taylor, 2010*).

The striatum is functionally connected to the threat detection system (*Haber, 2014*). Animal research has demonstrated that acute stressors including aversive stimuli induce a pronounced activation of the dopamine system in terms of dopamine neuron population activity (i.e. the numbers of neurons firing) and with regard to amphetamine-induced behaviours (*Valenti et al., 2011*). Long-lasting changes in dopamine function occur after single stress exposures, including altered responsivity to future stimulation (*Holly and Miczek, 2016*) in a manner similar to that induced by drugs of abuse (*Saal et al., 2003*) and such that stress plays a powerful role in the initiation, escalation, and relapse to drug abuse via dopaminergic mechanisms (*Koob and Volkow, 2016*).

In humans, childhood sexual abuse is associated with elevated urinary dopamine metabolites in childhood (*De Bellis et al., 1994*) and acute psychosocial stressors induce greater dopamine release in people with low self-reported maternal care (*Pruessner et al., 2004*). Stress-induced elevations in cortisol levels have been directly correlated with amphetamine-induced dopamine release (*Wand et al., 2007*) on the one hand, while corticotrophin-releasing hormone administration results in dopamine release on the other (*Payer et al., 2017*). In terms of long-term exposure, using fMRI in humans, there is evidence that institutional neglect is associated with reduced striatal reward function, which is mediated by the dopamine system (*Mehta et al., 2010*), and similar findings have been observed in prospective cohorts following childhood adversity (*Dillon et al., 2009*) suggestive of a possible causal relationship between childhood adversity and alterations of the dopamine system. Furthermore, maltreatment-associated reduced striatal function is associated with adverse outcomes including disrupted attachment (*Takiguchi et al., 2015*) and depression (*Hanson et al., 2015*). One study (*Oswald et al., 2014*) found positive associations between childhood trauma and amphetamine-induced dopamine release, which may be due to the phenomenon of cross-sensitization. Furthermore, within people who are at ultra-high clinical risk of psychosis raised dopamine synthesis capacity has been reported in those patients with high levels of childhood adversity (*Egerton et al., 2016*). In light of these findings, we wanted to examine the relationships between the autonomic, endocrine and subjective threat responses to an acute psychosocial stressor and dopaminergic function.

The relationships between neurobiological pathways and stress-induced physiological and subjective responses have yet to be fully elucidated in humans. Studies of psychosocial stressors and dopamine function have typically investigated risk factors in isolation, despite the fact that the risk factors cluster together and may share common underlying mechanisms (*Hjern et al., 2004*; *Morgan and Fisher, 2007*; *Wicks et al., 2005*). Since associations between one exposure and outcomes remain after controlling for the other exposures (*Schäfer and Fisher, 2011*), it is likely that additive effects and/or synergistic effects operate between risk factors (*Morgan et al., 2014*; *Guloksuz et al., 2015*; *Lataster et al., 2012*; *Morgan et al., 2008*; *Schäfer and Fisher, 2011*). Furthermore, there is evidence that childhood trauma increases risk of psychopathology in response to adult stressors (*McLaughlin et al., 2010*). It is also likely that ethnic minority status can increase the risk of psychopathology through social isolation, experiences of discrimination, victimisation and social defeat, which are all considered stressors (*Morgan et al., 2010*; *Bécares et al., 2009*; *Cooper et al., 2008*; *Cooper et al., 2017*; *Selten and Cantor-Graae, 2005*; *Selten et al., 2012*; *Sharpley et al., 2001*; *Tidey and Miczek, 1996*). Cognitive models propose that minority status is associated with greater levels of social threat (*Combs et al., 2002*; *Morgan and Fisher, 2007*). Indeed we have found evidence that black minority ethnic status is associated with greater amygdalar activation to out-group (i.e. white faces) than vice versa, neurobiological correlates of these theories/findings (*McCutcheon et al., 2018*) (*McCutcheon et al., 2018*). Taken with findings that experiences of racism are correlated with amygdala activation to white faces in black individuals (*Greer et al., 2012*),

this suggests that ethnic minority status is associated with functional alterations in the brain circuits involved in threat processing. As minority ethnicity status has been found to be a chronic stressor and increase the risk of mental illness via chronic stress rather than a genetic component (*Akdeniz et al., 2014*), we included minority ethnic status as a stressor. The combined effect of the risk factors on dopamine function in humans is unknown. Furthermore, previous studies (*Egerton et al., 2016*; *Oswald et al., 2014*; *Pruessner et al., 2004*) in humans have investigated childhood factors alone. Given animal evidence that exposure to mild stressors potentiate dopaminergic activity whilst severe chronic stressors is associated with dopaminergic blunting (*Holly and Miczek, 2016*), a key outstanding question remains - what is the effect of chronic adversity, across both child and adult stages of life, on dopaminergic function? We therefore aimed to investigate the effects of exposures to multiple psychosocial risk factors for psychosis on dopaminergic function and the acute stress response. Given the findings of dopaminergic dysfunction associated with childhood maltreatment presented above, we hypothesised that healthy humans with a high cumulative exposure to psychosocial stressors would have altered striatal dopamine synthesis, compared to humans with a low exposure. We also sought examine the relationship between dopaminergic function and the subjective threat and physiological responses to acute psychosocial stress using the Montréal Imaging Stress Task (MIST), a validated stress task involving mental arithmetic under negative social appraisal (*Dedovic et al., 2005*; *Lederbogen et al., 2011*). We sought to measure salivary α-amylase, secreted from the parotid gland in response to adrenergic activity and a marker of stress-induced adrenergic activity (*van Stegeren et al., 2006*) which is associated with a faster increase during psychosocial stress than salivary cortisol (*Maruyama et al., 2012*), and mean arterial pressure (MAP), the product of cardiac output and total peripheral resistance, reflecting organ perfusion and providing a physiological measure of sympathetic activation.

## Results

### Participant characteristics and scan parameters

Seventeen HA participants were recruited to the study. All reported high levels of psychological stress exposure in childhood and adulthood (*Table 1*). Seventeen LA participants were recruited and, as expected, had significantly lower levels of childhood and adult stressors than the HA group (*Table 1*). Clinical rating scales are reported in *Table 1*. HA scored significantly higher than LA on subclinical measures of depressive symptoms (BDI), the degree to which previous stressors were having an impact on their lives in the week prior to scanning (BIE) and aberrant salience (ASI).

There was no significant group difference in the amount of radioactivity injected or specific activity (*Table 1*). There was no significant difference in whole striatal or subdivision volumes between the groups (*Table 1*).

### Striatal dopaminergic function

$K_i^{cer}$ was significantly reduced in HA relative to LA in the whole striatum ($t_{32}$ = 2.27, p=0.03; *Figure 1*). Secondary analysis in each striatal subdivision showed that this reduction reached significance in the limbic and associative subdivisions (*Table 2*).

As the amount of smoking differed in the groups and heavy smoking can influence dopamine function (*Bloomfield et al., 2014*; *Salokangas et al., 2000*), we performed an ANCOVA to examine whether smoking was influencing our findings. When co-varying for amount of current cigarette use, the group difference remained significant in the limbic striatum only ( $F_{1,30}$ = 5.2, p=0.029, $\eta^2 p$=0.15).

### Psychosocial stress-induced effects

There were no differences between the groups on subjective or biological measures of stress response at baseline (*Table 3*) apart from a statistical trend (p=0.07, *d* = 0.67) towards greater amylase concentrations in the HA group compared to the LA group. Response to psychosocial stress is shown in *Table 4* and *Figure 2*. The HA group showed a heightened subjective response to psychosocial stress evidenced by Threatened (p=0.04, *d* = 0.86) compared to the LA group. By contrast, the HA showed a blunted physiological response to psychosocial stress, as evidenced by an attenuated increase in mean arterial blood pressure (MAP), p=0.03, *d* = 0.81 and trend for a

**Table 1.** Sample characteristics and scan parameters

| Sample characteristic | LA ( = 17) | | HA ( = 17) | | pª |
|---|---|---|---|---|---|
| Age, years [mean(SD)] | 27.6 | (7.8) | 29.2 | (7.2) | 0.54 |
| Sex, n | nine female, eight male | | eight female, nine male | | 1.00 |
| Ethnicity, n | 17 WB | | 4 BA, 1BB, 4 BC, 6 ME, 1 OE, 1 WB | | <0.001 |
| *Childhood Adversity* | | | | | |
| CTQ [mean(SD)] | 3.8 | (5.2) | 15.3 | (16.1) | 0.01 |
| Parental loss (parental separation with loss of parental contact and/or death and/or going into foster care and/or being adopted) during childhood, n | 0 | | 13 | | <0.001 |
| Childhood sexual abuse | 0 | | 6 | | 0.02 |
| *Adult Adversity* | | | | | |
| Number of adverse life events over last 6 months [mean(SD)] | 0.5 (0.9) | | 2.6 (1.9) | | 0.001 |
| Life events score over last 6 months [mean(SD)] | 15.1 (37.0) | | 72.3 (55.7) | | <0.01 |
| *Clinical Scores* | | | | | |
| BDI [mean(SD)] | 2.7 (3.8) | | 6.5 (5.6) | | 0.03 |
| BAI [mean(SD)] | 4.8 (6.7) | | 9.7 (10.2) | | 0.11 |
| IES-6 [mean(SD)] | 1.7 (2.3) | | 7.7 (7.6) | | 0.01 |
| O-LIFE [mean(SD)] | 7.2 (6.5) | | 13.1 (9.5) | | 0.07 |
| ASI [mean(SD)] | 5.7 (5.8) | | 11.6 (7.5) | | 0.02 |
| *Current Drug Use*[c,d] | | | | | |
| Tobacco cigarette smokers in last 3 months (n) | three user, 14 non-users | | four users, 13 non-users | | 1.00 |
| Tobacco use in whole sample (cigarettes/day) [mean(SD)] | .4 | (1.5) | 1.7 | (3.6) | 0.19 |
| Alcohol use in last 3 months (n) | 15 users, two non-users | | 14 users, three non-users | | 1.00 |
| Alcohol use (UK alcohol units/week) [mean(SD)] | 10.2 | (9.0) | 7.0 | (8.9) | 0.30 |
| *Scan parameter* | | | | | |
| Injected dose (MBq) [mean(SD)] | 143.4 | (7.7) | 142.9 | (7.7) | 0.85 |
| Specific activity (MBq/µmol) [mean(SD)] | 35.3 | (6.7) | 41.4 | (15.4) | 0.14 |
| Whole striatal volume (mm³) [mean(SD)] | 16,842 | (5094) | 15,741 | (4,601) | 0.54 |
| Associative striatal volume (mm³) [mean(SD)] | 10,460 | (3202) | 9771 | (2885) | 0.54 |
| Limbic striatal volume (mm³) [mean(SD)] | 2005 | (610) | 1897 | (547) | 0.61 |
| Sensorimotor striatal volume (mm³) [mean(SD)] | 4375 | (1314) | 4072 | (1189) | 0.51 |

Abbreviations: ASI, Aberrant Salience Inventory; BA, black African; BAI; Beck Anxiety Inventory; BB, black British; BC, black Caribbean; BDI, Beck Depression Inventory; CTQ, Childhood Trauma Questionnaire; IES-6, Brief Impact of Events Scale;, mixed ethnicity; OE, other ethnicity; O-LIFE, Oxford-Liverpool Inventory of Feelings and Experiences; SEAT, Social Environment Assessment Tool; WB, White British.

[a] Independent-samples *t*-tests for variables with normal data distributions; Mann-Whitney U tests for variables with non-normal data distributions; $\chi^2$-tests for dichotomous variables.
[b] Groups were compared on a dichotomised ethnicity variable (white British vs ethnic minority).
[c] 1 UK alcohol unit = 10 mL (~7.88 g) alcohol.
DOI: https://doi.org/10.7554/eLife.46797.002

lower increase in cortisol (p=0.06, *d* = 0.69). Some participants had a negative area under the curve due a reduction in salivary cortisol levels associated with the task.

## The relationships between physiological and subjective measures

We conducted correlations between the primary outcome of interest (dopamine synthesis capacity in the whole striatum) and variables showing differences in response to acute psychosocial stress (AUC for threatened, cortisol and mean arterial blood pressure). Extreme bivariate outliers (Cook's distance >1) were removed (*n* = 7 from threat, *n* = 2 from cortisol, *n* = 5 for MAP). In the low adversity group, striatal dopamine synthesis capacity correlated with psychosocial stress-induced threat (r = 0.73, p=0.001, *Figure 3A*) and mean arterial blood pressure (r = −0.62, p=0.013, *Figure 3B*). There were no correlations between striatal dopamine synthesis capacity and measures of acute psychosocial response in the high adversity group.

As we found that dopamine synthesis capacity is correlated with both subjective and physiological response to stress in LA, we performed a regression analysis to identify which subregion is the best predictor for the physiological measures. Regression analyses did not identify which striatal subregion was the best predictor of the physiological measures (p>0.08).

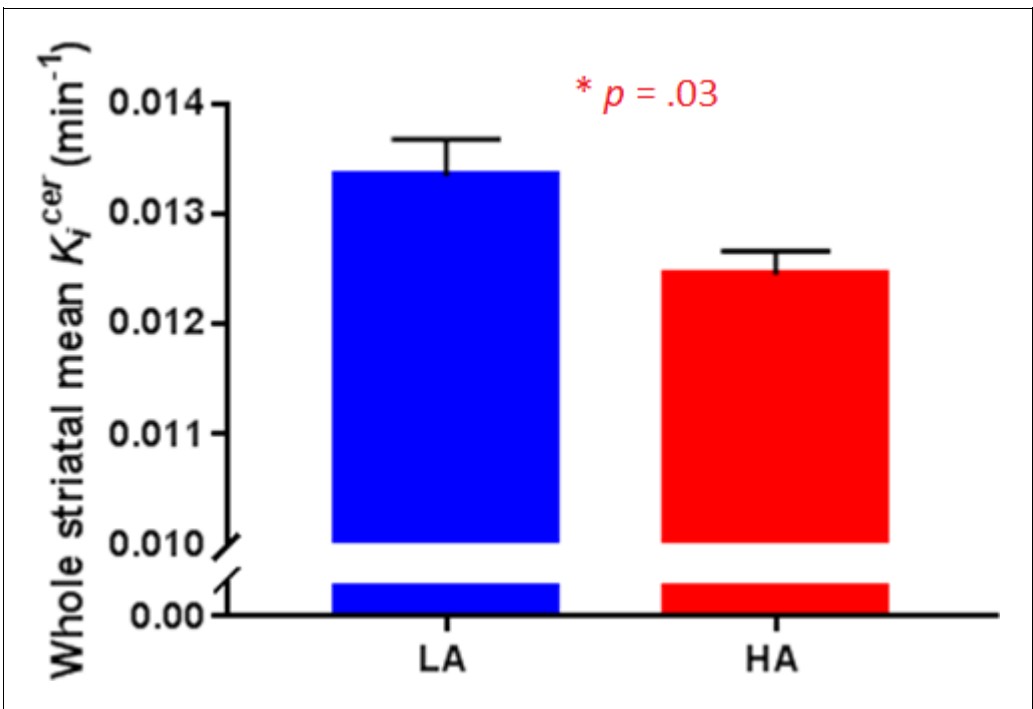

**Figure 1.** Striatal dopamine synthesis capacity in Low Adversity (LA, n = 17) and High Adversity participants (HA, n = 17). Dopamine synthesis capacity was significantly reduced in HA compared with LA ($t_{32}$ = 2.27, p=0.03). Error bars indicate standard errors.
DOI: https://doi.org/10.7554/eLife.46797.003

**Table 2.** [$^{18}$F]-DOPA $K_i^{cer}$ (min$^{-1}$) by group

| VOI | LA ( = 17) | | HA ( = 17) | | Group comparisons[a] | | Effect size |
|---|---|---|---|---|---|---|---|
| | Mean | (SD) | Mean | (SD) | $t_{df}$ | p | (Cohen's D) |
| STR | 0.0133 | (0.0014) | 0.0124 | (0.0013) | $2.27_{32}$ | 0.03 | 0.80 |
| AST | 0.0133 | (0.0011) | 0.0124 | (0.0010) | $2.28_{32}$ | 0.03 | 0.81 |
| LST | 0.0140 | (0.0015) | 0.0128 | (0.0010) | $2.69_{32}$ | 0.01 | 0.95 |
| SMST | 0.0132 | (0.0013) | 0.0125 | (0.0011) | $1.17_{32}$ | 0.10 | 0.41 |

Abbreviations: AST, associative striatum; LST, limbic striatum; $K_i^{cer}$, influx rate constant; SMST, sensorimotor striatum; STR, whole striatum; VOI, volume of interest.

[a] Independent-samples $t$-tests.

DOI: https://doi.org/10.7554/eLife.46797.004

## Discussion

Our main finding is that chronic exposure to psychosocial stressors is associated with significantly reduced striatal dopamine synthesis capacity, particularly in the limbic (ventral) striatum. In addition, we found evidence that striatal dopamine synthesis capacity correlated with both the physiological and subjective responses to an acute psychosocial stressor. Chronic stress exposure is associated with a dissociation between physiological and psychological acute stress responses in the form of an attenuated stress-induced increase in blood pressure alongside a potentiated stress-induced subjective response. These findings support our hypothesis that high cumulative exposure to psychosocial adversity would be associated with altered dopamine synthesis capacity.

### Interpretation of findings

Acute stress is associated with increased dopaminergic and autonomic output in animals and humans (*Imperato et al., 1989*; *Wand et al., 2007*) and recent human evidence indicates that corticotrophin-releasing hormone administration results in dopamine release (*Payer et al., 2017*). Animal research indicates that dopaminergic neurons are strongly excited by acute stress and aversive stimuli during adulthood (*Brischoux et al., 2009*; *Cohen et al., 2012*; *Wenzel et al., 2015*; *Zweifel et al., 2011*). Long-lasting neuroadaptive changes on VTA dopamine neurons have been observed after a single stress exposure, demonstrating that acute stress can alter VTA dopamine neuron responsivity to future stimulation (*Saal et al., 2003*). Exposure to a single acute stressor can also promote long-lasting neuroplastic changes in VTA dopamine neurons in a manner similar to exposure to recreational drugs (*Dong et al., 2004*; *Graziane et al., 2013*; *Niehaus et al., 2010*; *Saal et al., 2003*). Our results extend these findings and are consistent with evidence from animal models whereby subcortical dopamine transmission is blunted in response to multiple stressors in adulthood (*Chrapusta et al., 1997*; *Gresch et al., 1994*).

Whilst stress exposure in animals during the juvenile period and adolescence has a very different effect from to chronic stress in adulthood, our findings are also broadly consistent with developmental stress models (*Brake et al., 2004*; *Meaney et al., 2002*). Likewise, early maternal deprivation

**Table 3.** Baseline stress reactivity in Low Adversity (LA) and High Adversity (HA) groups at prior to acute psychosocial stress challenge

| Measure | LA ( = 17) | | HA ( = 17) | | Group comparisons | | Effect size |
|---|---|---|---|---|---|---|---|
| | Mean | (SD) | Mean | (SD) | $t_{df}$ | P | (Cohen's D) |
| Threatened (mm) | 6.75 | (15.52) | 4.58 | (6.00) | $0.46_{26}$ | 0.65 | 0.18 |
| Cortisol (U/mL) | 3.93 | (2.74) | 5.12 | (3.54) | $1.08_{30}$ | 0.29 | 0.38 |
| Amylase (U/mL) | 178.42 | (173.83) | 92.79 | (53.13) | $1.83_{30}$ | 0.07 | 0.67 |
| MAP (mmHg) | 89.67 | (9.45) | 90.38 | (9.63) | $0.21_{29}$ | 0.84 | 0.07 |

Abbreviations: HA, high adversity; LA, low adversity; MAP, mean arterial pressure.

DOI: https://doi.org/10.7554/eLife.46797.005

**Table 4.** Acute response to psychosocial stress challenge in Low Adversity (LA) and High Adversity (HA) groups

| Measure | LA ( = 17) | | HA ( = 17) | | Group comparisons | | Effect size |
|---|---|---|---|---|---|---|---|
| | Mean | (SD) | Mean | (SD) | $t_{df}$ | p | (Cohen's D) |
| Threatened (AUC) | 191.25 | (587.99) | 780.83 | (764.33) | $2.31_{26}$ | 0.04 | 0.86 |
| Cortisol (AUC) | 122.34 | (156.49) | 11.75 | (166.14) | $1.94_{30}$ | 0.06 | 0.69 |
| Amylase (AUC) | 1616.67 | (5750.66) | 1015.84 | (2740.24) | $0.37_{30}$ | 0.72 | 0.13 |
| MAP (AUC) | 153.30 | (90.04) | 79.31 | (92.09) | $2.26_{29}$ | 0.03 | 0.81 |

Abbreviations: HA, high adversity; LA, low adversity; MAP, mean arterial pressure.

DOI: https://doi.org/10.7554/eLife.46797.006

models in the very early juvenile period (from post-natal day 5) have been associated with hypodo-paminergic behaviours in later life including reduced or attenuated responses to acute stress, conditioned locomotion, locomotor activity, and dopamine agonist-induced locomotion with amphetamine, alongside potentiated dopamine antagonist-induced decreases in anticipatory responses (*Matthews et al., 1996*; *Hall et al., 1999*). Exposure to repeated longer-term stressors leads to decreases in striatal dopamine function including nucleus accumbens dopamine output (*Mangiavacchi et al., 2001*), reduced cocaine-induced nucleus accumbens dopamine release (*Shimamoto et al., 2011*; *Holly and Miczek, 2016*), and reduced striatal dopamine receptor availability (*Meaney et al., 2002*; *Brake et al., 2004*). One explanation for the group difference is that long-term exposure to psychosocial stress is associated with dopaminergic, particularly in the limbic striatum, autonomic and endocrine downregulation. There is evidence of regional specificity in the direction of effects of acute vs chronic stress exposure. Acute and repeated stress activates the entire dopamine system projecting to much of the striatum (*Valenti et al., 2011*), in particular the associative (dorsal) striatum where object salience is important, whereas in chronic stress-induced depression (*Holly and Miczek, 2016*), the blunting occurs primarily in the neurons projecting to the ventromedial striatum (*Moreines et al., 2017*), where reward-related variables are processed. These are therefore likely to be different systems mediating the dopamine stress response that varies with duration of stress exposure, and with the induction of anxiety (acute or repeated stress) vs. depression (chronic stress). Our findings are consistent with Koob's opponent process model, where acute stress activates the dopamine system, which upon chronic exposure leads to a compensatory downregulation (*Koob et al., 1997*).

However, this interpretation is not consistent with all findings (*Butterfield et al., 1999*; *Tidey and Miczek, 1996*). These discrepant findings are likely to reflect differences in the stress paradigm employed as mild stressors tend to potentiate dopamine function and severe/chronic stressors tend to reduce activity (*Holly and Miczek, 2016*), which could be consistent with an adaptive role for dopamine in response to mild stressors, but chronic uncontrollable stress hijacking this system. For example, maternal deprivation and isolation of neonatal rats was associated with increased dopamine release (*Hall et al., 1999*; *Kosten et al., 2003*), whilst unavoidable stress administered over one week and three weeks was associated with a decrease in nucleus accumbens dopamine output (*Mangiavacchi et al., 2001*). Likewise, rats under a 10-day episodic defeat paradigm had a sensitised dopamine response in the nucleus accumbens, whilst when under a 5-week continuous subordination paradigm they exhibited a suppressed dopamine response (*Miczek et al., 2011*).

Our findings are consistent with a fMRI study with found that adolescents who had suffered severe early life deprivation exhibited ventral striatal hyporesponsivity during anticipation of monetary reward (*Mehta et al., 2010*). Yet, findings of increased ventral striatal dopamine release in response to psychosocial stress in humans who reported insufficient early life maternal care (*Pruessner et al., 2004*), and positive associations between childhood adversity and amphetamine-induced dopamine release are not consistent with these results (*Oswald et al., 2014*). Since the HA group reported high levels of adverse psychosocial experiences throughout their lives, it is therefore possible that exposure to moderate stressors results in an initial sensitisation of dopaminergic function whereas repeated exposures to severe stressors result in a subsequent down-regulation. Alternatively, it is possible that early life stress is non-linearly related to the responsivity of the cortisol and dopamine systems (*Del Giudice et al., 2011*). Other factors such as type of stressor may also cause different dopaminergic, autonomic and endocrine effects as work to date has demonstrated

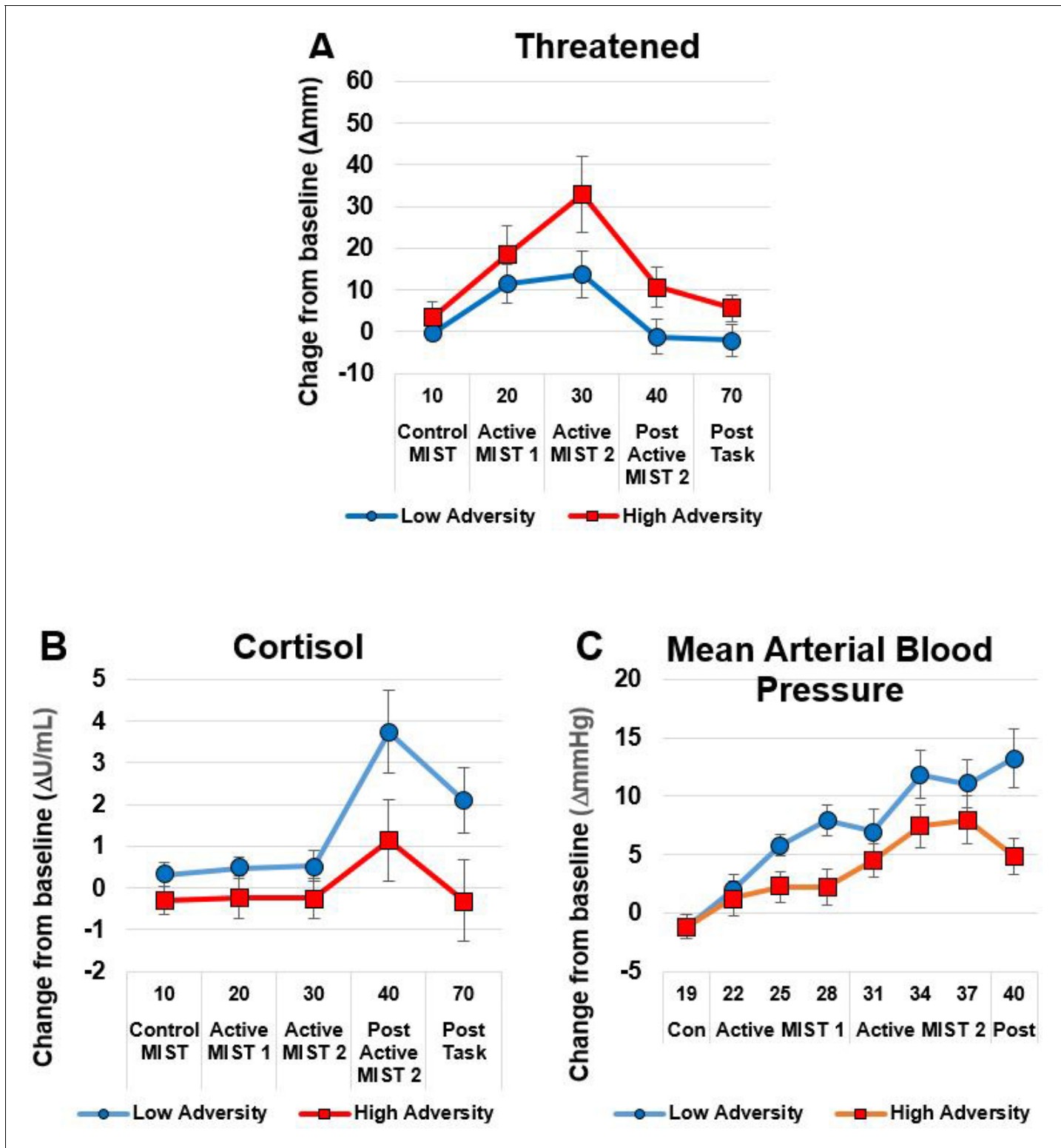

**Figure 2.** The High Adversity group showed a heightened subjective response and a blunted physiological response. Panel A shows subjective Threatened responses; Panels B (Cortisol) and C (Mean Arterial Blood Pressure) show physiological response. Data show mean (+ /- SEM).
DOI: https://doi.org/10.7554/eLife.46797.007

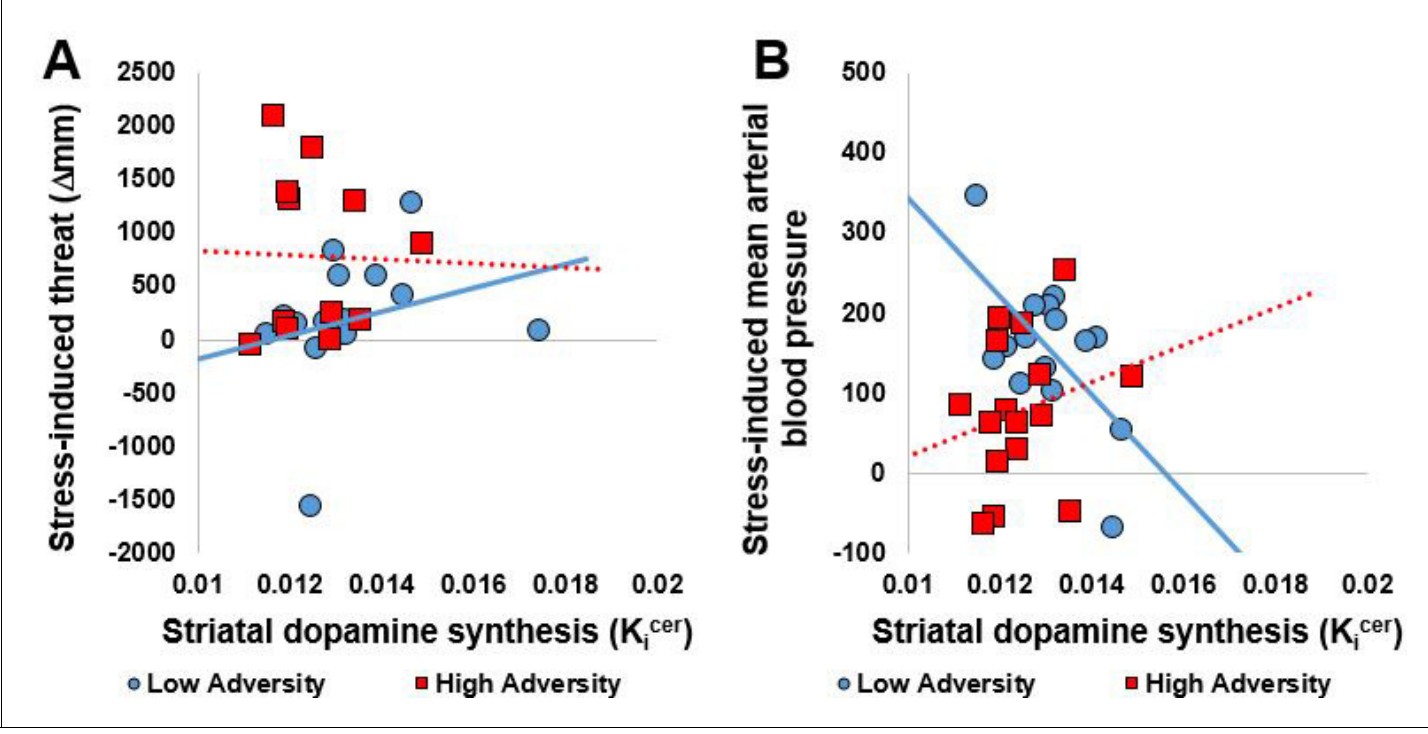

**Figure 3.** Correlations between striatal dopamine synthesis capacity and acute response to psychosocial stress. (A) Whole striatal dopamine synthesis capacity was positively correlated with stress-induced threat in the Low Adversity group ($r$ = 0.73, p=0.001) but not the High Adversity group ($r$ = −0.03, p=0.934). (B). Whole striatal dopamine synthesis capacity was negatively correlated with stress-induced threat and mean arterial blood pressure in the Low Adversity group ($r$ = −0.62, p=0.013) but not the High Adversity group ($r$ = 0.23, p=0.395). Extreme bivariate outliers have been removed from the figures.

DOI: https://doi.org/10.7554/eLife.46797.008

that different schedules, intensities, or modalities of stressor presentation can result in dramatically different behavioural and physiological responses (*Holly and Miczek, 2016*) and these stressor-specific effects are appear highly region specific. Taken together with findings of stress-induced increases in dopaminergic measures (e.g. *Fulford and Marsden, 1998*; *Hall et al., 1999*; *Kosten et al., 2003*; *Tidey and Miczek, 1996*), the nature, intensity, and schedule of repeated stress may be critical, such that mild or intermittent stressors appear to potentiate basal VTA dopamine neuron activity and more severe or chronic uncontrollable stressors appear to reduce basal VTA dopamine activity, and the response to later stressors of a different nature is generally cross-sensitized (*Holly and Miczek, 2016*). An alternative explanation for the seemingly discrepant finding is that different parts of dopaminergic system undergo divergent responses to repeated stress (*Brake et al., 2004*). Additionally, differences in stress-induced dopaminergic outputs have even been observed within the same single neuron projections depending on the location of glutamatergic inputs (*Finlay and Zigmond, 1997*).

Whilst our measures of stress-induced change in endocrine and physiological function returned to basal levels, we cannot exclude the possibility that inducing acute psychosocial stress prior to PET scanning had an effect on the result. Few studies have measured dopamine synthesis in the period following an acute stressor. One study, for example, found that dopamine synthesis was reduced following restraint stress (*Demarest et al., 1985*) which may have been due to the activation of inhibitory feedback mechanisms. It therefore remains possible, albeit unlikely, that our findings are due to acute up-regulation of inhibitory feedback mechanisms including stress-induced dopamine release inhibiting dopamine synthesis via autoreceptors (*Castro et al., 1996*). Alternatively, since corticosteroids regulate tyrosine hydroxylase activity (the rate-limiting step in the dopamine synthesis pathway) (*Meyer, 1985*), and there is evidence that acute corticosteroids are associated with subsequent decreased striatal dopamine synthesis (*Lindley et al., 1999*), it is possible, therefore, that our

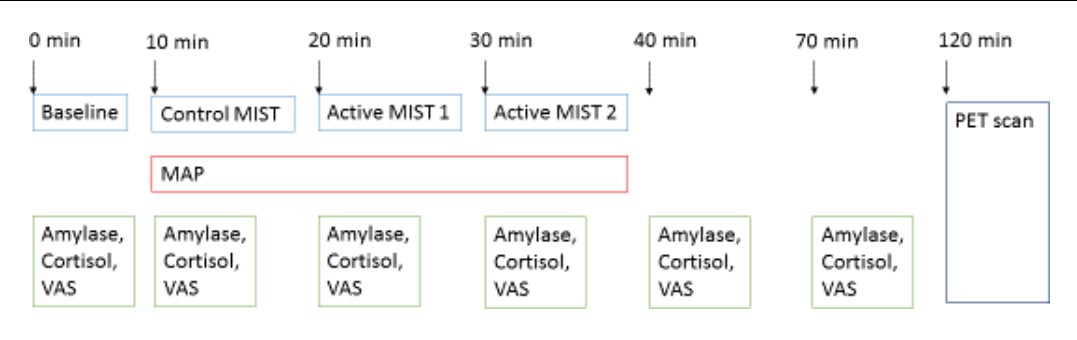

**Figure 4.** Experimental procedures. MIST, Montreal Imaging Stress Test; MAP, Mean Arterial Blood Pressure; VAS, visual analogue scale.
DOI: https://doi.org/10.7554/eLife.46797.009

findings are due to participants with chronic stress being sensitized to the acute stress-induced decreases in dopamine synthesis via corticosteroid mediated pathways.

Coherence between the emotional, endocrine and autonomic stress outcome systems has been assumed by some but has been questioned by others (*Campbell and Ehlert, 2012*; *Mauss et al., 2005*). Thus, our findings of divergent physiological and subjective emotional responses associated with chronic stressors in the current study are therefore of interest to the field. Repeated exposure to corticosteroids can lead to attenuation of HPA axis activity through negative feedback mechanisms (*Karssen et al., 2005*) and so it is not surprising that repeated exposure to stressors would be associated with a reduced autonomic response to acute stress, as has been found in adolescent survivors of developmental trauma (*Gooding et al., 2016*). Propranolol and dexamethasone attenuate the autonomic and endocrine responses to acute psychosocial stress but not the psychological response (*Ali et al., 2017*) indicating that these responses can be dissociated. Nonetheless, the mechanisms underlying the dissociation between these measures and subjective emotions are unclear and further research is needed to elucidate these dissociative mechanisms. One possibility is that this dissociation reflects a shift away from dopaminergic mechanisms to a more cortisol based stress response. Our findings of exposure to stressors being associated with autonomic effects are also important in the context of understanding the mechanisms underlying the well-established associations between cardiovascular disease and both depression (*Penninx, 2017*) and psychosis (*Osborn et al., 2007*).

Our findings of heightened stress-induced threat are consistent with the threat-anticipation model of delusion formation (*Freeman, 2007*) and evidence that elevated sensitivity to socio-environmental stress via enhanced threat anticipation in daily life may be important psychological processes underlying the association between childhood adversity and psychosis (*Reininghaus et al., 2016*). Given our findings of decreased dopamine synthesis capacity associated with high psychosocial stressors, it remains possible that repeated psychosocial stressors in childhood increase the risk of psychotic symptoms by hypodopaminergic or non-dopaminergic processes. However, *Thompson et al. (2013)* found that patients with comorbid schizophrenia and substance dependence (associated with a blunted dopamine system) had reduced amphetamine-induced dopamine release. Yet, despite a blunted dopamine response, this study found the classically described relationship between dopamine release and increase in psychotic symptoms, which may be due to $D_2$ receptor super-sensitivity (*Seeman and Seeman, 2014*). Alternatively, as these participants were healthy it remains possible that these participants have resilience and so the observed findings may be due to adaptive allostatic down-regulations (*McEwen, 1998*).

Although the case–control design of this study is not able to confirm a causative relationship between psychosocial stress and dopamine dysfunction, these findings warrant further research into potential causative mechanisms. Our findings, particularly of relationships between cumulative stress exposure and the dopamine system, may be important for understanding how exposures to multiple stressors induce changes in the dopamine system, and how these relate to both vulnerability to and resilience against mental illnesses. It would be important to consider these findings in light of a putative role of the dopaminergic system in the pathophysiology anhedonia in depression (*Nutt, 2006*),

the role of dopamine in social motivation (*Love, 2014*), and findings that striatal neurons can incorporate social reward into their computations (*Báez-Mendoza et al., 2013*; *Schultz, 2016*). Our findings may be highly relevant in terms of our understanding of addiction, as a history of exposure to aversive stimuli is strongly associated with later addictive behaviour, with both clinical and preclinical work demonstrating that stress plays a powerful role in the initiation, escalation, and relapse to drug abuse (*Shaham et al., 2000*; *Sinha, 2007*; *Sinha, 2009*).

## Strengths and limitations

A strength of our study is that it specifically examined the effect of multiple psychosocial risk factors to examine their combined effect since these factors often cluster together in the general population (*Hjern et al., 2004*; *Teicher et al., 2016*; *Wicks et al., 2005*) and so it is not possible to disentangle the different types of psychosocial adversity due to lack of power as we are unable to directly contrast the effect of single risk factors with multiple exposures, or determine if risk factors have synergistic effects. Likewise, we did not co-vary for the different psychosocial risks as analysis of covariance is suboptimal at adjusting for factors when groups differ significantly in their covariates (*Miller and Chapman, 2001*). We chose to recruit participants with high levels of stress exposure in early development and adulthood, because early developmental stressors increase the risk of psychopathology following adult stressors (*McLaughlin et al., 2010*). However, a potential limitation of our combination of early developmental and adult stressors is that it may confound the early life (i.e. likely programming) effects with the later in life acute stressors which occur after developmental sensitive periods.

The PET scan was conducted following exposure to acute psychosocial stress exposure. We sought to reduce the length of time between task and PET scan to reduce variance in time and to limit the variance associated with normal fluctuations in dopamine synthesis capacity that is to limit the temporal separation between the stress task and the PET scan. Whilst acute stress increases dopamine release (*Pruessner et al., 2004*), it is not clear if this has an acute effect on striatal dopamine synthesis capacity. Therefore, it is possible that the close temporal proximity of the stress task to the PET means that what was observed during the PET scan could have been influenced by the stress task. Thus, one possible explanation for our findings could be reduced change in dopamine synthesis capacity following stress in the high adversity group relative to the low stress group, which would be consistent with Koob's model (*Koob et al., 1997*). However, we are not aware of evidence indicating that an acute stressor can alter the parameters that contribute to our index (i.e. the activity of aromatic acid decarboxylase) in the timeframe used in our study. Nonetheless, a controlled study comparing stress and no stress prior to PET to investigate the effects is needed to definitively determine if acute stress alters dopamine synthesis capacity. Given this, it is possible that an optimal experimental design would allow more time to pass between the stressor and the PET scan.

A further consideration is that our groups were not matched for ethnicity. Ethnicity is associated with differences in allele frequencies of dopaminergic genetic variants (*Gelernter et al., 1993*) and striatal $D_{2/3}$ receptor availability (*Wiers et al., 2018*) and so we cannot exclude the possibility that this is driving our results. Nonetheless, the study by Wiers and colleagues (*Wiers et al., 2018*) found ethnicity was associated with dorsal (rather than ventral) striatal dopamine receptor effects and that the effects of ethnicity on dopamine receptor availability were not driven by variation in dopamine candidate genes suggesting that their results are influenced by socioeconomic factors and therefore psychosocial stressors per se. Since our effects were most pronounced in the ventral (rather than dorsal) striatum and we found relationships between psychosocial stressors and ventral striatal dopamine synthesis, it is therefore unlikely that our results can be explained by ethnicity. Our participants were of multiple ethnic groups and so we were unable to determine the effects of psychosocial stress associated with immigration accounting for ethnicity. Future studies of individual risk factors are needed to examine what type of risks may be driving our findings. Finally, as our study is cross-sectional we cannot rule out reverse causality and a longitudinal design is required to distinguish between the interpretations discussed above.

We cannot exclude the possibility that recall bias may be confounding our findings as the measures of psychosocial stress rely on self-report and it was therefore not possible to independently verify the histories of psychosocial stressors. As such, the assessment of childhood trauma may be liable to recall bias in depressed patients (*Lewinsohn and Rosenbaum, 1987*). However, measures of childhood trauma have been demonstrated to remain stable over time and to be independent of the

current degree of abuse-related psychopathology (*Paivio, 2001*). Despite ongoing concerns that retrospective reporting overestimates associations between abuse and adult psychopathology (*Gilbert et al., 2009*), there is evidence that prospective and retrospective measures of abuse predict similar rates of mental illness (*Scott et al., 2012*), recall bias accounts for less than 1% of variance in measures of child abuse (*Fergusson et al., 2011*) and good reliability and validity has been reported for retrospective self-reports of early experiences obtained from individuals with psychotic disorder and so this is unlikely to be significantly confounding our results (*Fisher et al., 2011*). Nonetheless, difficulties remain in measuring and quantifying emotional neglect due in part to its highly subjective nature (*Watson et al., 2014*). We did not control for genetic influences, apart from excluding individuals with a family history of psychosis and so genetic differences between exposed and unexposed groups might contribute to our finding. Nonetheless, previous work in our laboratory (*Stokes et al., 2013*) on heritability of striatal dopamine synthesis capacity found that individual-specific environmental factors, rather than genetic factors, had the greatest effect on the limbic striatum which is consistent with the interpretation that the psychosocial exposures have contributed to our finding of striatal hypodopaminergia. A further limitation is that we did not account for menstrual cycle phase in our analysis, given effects on the HPA axis (*Kirschbaum et al., 1999*).

These findings show that long-term exposure to psychosocial stressors is associated with reduced striatal dopamine synthesis capacity, particularly in the limbic subdivision of the striatum, alongside a de-coupling of the acute stress response such that emotional responses are potentiated whilst cardiovascular and endocrine responses tended to be blunted. Further work is needed to understand what processes contribute to this decoupling and how this may contribute to the development of mental illness.

## Materials and methods

This study was approved by the National Research Ethics Service (12/LO/1955) and the Administration of Radioactive Substances Advisory Committee (ARSAC). The study was conducted in accordance with the Helsinki Declaration. All participants provided informed written consent to participate after an oral and written explanation of the study.

### Participant recruitment

We recruited two groups of healthy volunteers, one exposed to multiple risk factors (exposed high adversity group, 'HA')) and one not exposed (unexposed low adversity group, 'LA'), from throughout the UK via public advertisement, newspaper advertisement and national media engagement. Responding individuals were then screened via telephone. LA 'controls' were individually matched to the HA group on the basis of age (+ /- 5 years) and sex. Inclusion criteria for all participants were: age 18–45 years, good physical health and capacity to give written informed consent. Exclusion criteria for all participants included: a personal history of psychiatric illness including substance abuse but not Nicotine Use Disorders; a history of psychotic illness in first degree relatives; evidence of an at risk mental state (*Yung et al., 2005*) and contraindications to PET including pregnancy, nursing mothers, severe obesity and previous clinical procedures involving exposure to significant ionizing radiation within the last year.

Additional inclusion criteria for the HA (exposed) group included at least one childhood stressor and at least two adult stressors. These were ascertained by structured clinical interview. Childhood trauma self-reports were triangulated with the Childhood Trauma Questionnaire (CTQ) (*Bernstein and Fink, 1998*) and the Childhood Experiences of Care and Abuse (CECA) (*Bifulco et al., 2005*).

1. **Childhood stressors:** Childhood (before age 16 years); History of childhood (before age 16 years) adversity defined as one or more of the following: parental loss (including separation with loss of parental contact and/or death and/or going into foster care and/or being adopted) and/or abuse (including physical, sexual abuse or neglect) and/or bullying (i.e. peer abuse) and/or major disaster and/or war and/or admission to hospital with life-threatening medical problem.
2. **Adult stressor:** Minority ethnic status; a significant life event defined as a bereavement, moving house, a change in job or financial circumstances, a new family member being born, a breakdown of a significant relationship, and/or unemployment within the last six months.

Additional inclusion criteria for the LA group included no exposure to the childhood factors listed above, ethnic majority status and no significant adverse life events in the last 6 months.

## Psychosocial assessments

Assessments included the Beck Depression Inventory (BDI; *Beck et al., 1996*) Beck Anxiety Inventory (BAI) (*Beck et al., 1988*), Aberrant Salience Inventory (*Cicero et al., 2010*), the CTQ and an adapted bullying questionnaire (*Olweus, 1996*). Detailed histories of life events over the preceding 6 months were obtained via the List of Threatening Events (*Brugha and Cragg, 1990*), and a life events score then calculated from these events based on the Holmes and Rahe life events stress scale (*Holmes and Rahe, 1967*); Brief Impact of Events Scale (IES-6) (*Thoresen et al., 2010*).

## PET scans

Participants were asked to fast for 5 hr and to refrain from smoking tobacco for 2 hr before imaging. On the day of the PET scan, urine drug screen (Monitect HC12, Branan Medical Corporation, Irvine, California) confirmed no recent drug use, and a negative urinary pregnancy test was required in all female participants. Head position was marked and monitored via laser crosshairs and video camera, and minimized using a head-strap. We used a Siemens Biograph 6 TruePoint PET-CT scanner (Siemens Healthcare, Erlangen, Germany). A computed tomography (CT) scan (effective dose = 0.36 mSv) was acquired for attenuation and model-based scatter correction prior to each PET scan. A target dose of approximately 150 MBq of [$^{18}$F]-DOPA was administered by bolus intravenous injection at the start of PET imaging. Emission data were acquired in list mode for 95 min, reconstructed in a 128 × 128 matrix with 2.6x zoom via filter back projection with a three dimensional 5 mm full-width half-maximum Gaussian image filter and re-binned into 32 timeframes (comprising eight 15 s frames, three 60 s frames, five 120 s frames, and sixteen 300 s frames).

## Image analysis

To correct for head movement in the scanner, non-attenuation-corrected dynamic images were denoised using a level 2, order 64 Battle-Lemarie wavelet filter. Nonattenuation-corrected images were used for the realignment algorithm as they include greater scalp signal, improving re-alignment compared with attenuated-corrected images (*Turkheimer et al., 1999*). Frames were realigned to a single 'reference' frame, acquired 10 min post-injection, employing a mutual information algorithm (*Studholme et al., 1996*). The transformation parameters were then applied to the corresponding attenuated-corrected dynamic images. The realigned frames were then summated, creating a movement-corrected dynamic image, which was used in the analysis. The cerebellar reference region (*Kumakura and Cumming, 2009*) was defined using a probabilistic atlas (*Martinez et al., 2003*), and as previously described, regions of interest (ROI) in the whole striatum and its functional subdivisions (*Haber, 2014*) were delineated to create an ROI map (*Egerton et al., 2010*). SPM8 (http://www.fil.ion.ucl.ac.uk/spm) was then used to normalize the ROI map together with the tracer-specific ([$^{18}$F]-DOPA) template (*Egerton et al., 2016*; *Howes et al., 2009*) to each individual PET summation image. This nonlinear transformation procedure allowed ROIs to be automatically placed on individual [$^{18}$F]-DOPA PET dynamic images. The influx rate constants ($K_i^{cer}$, written as $K_i$ in some previous publications (*Howes et al., 2013*) for the entire striatal ROI and the functional subdivisions bilaterally were calculated compared with uptake in the reference region using a graphical approach adapted for a reference tissue input function (*Egerton et al., 2016*).

## Psychosocial stress paradigm

We induced psychosocial stress using the Montreal Imaging Stress Task (MIST) (*Pruessner et al., 2004*) 2 hr before PET scanning. The rationale for conducting the PET scan on the same day as the stress task was to reduce the variance in the time between the measures. Participants were aware that one of the measures would be inducing psychosocial stress; however, they were only told after the MIST was completed that this was the task to induce psychosocial stress, and they were then debriefed. They were told beforehand that they could stop the experimental procedures at any time. During the MIST, participants were asked to solve mental arithmetic problems first under a control condition during which no time constraint or feedback were present, and subsequently under

the experimental condition where time and difficulty were automatically adjusted to result in a 30–40% error rate. During the experimental condition, we continuously made participants aware of their suboptimal performance via a visual performance bar and scripted verbal negative feedback delivered approximately every 1 min, where a confederate researcher reminded participants that they were performing much worse than average. The MIST was administered using pairs of researchers who were balanced for sex and ethnicity (i.e. male and female researchers, white British and minority ethnicities). There were 2 × 4 min blocks of control MIST control followed by a brief rest, and then 2 pairs of 2 × 4 min blocks of the experimental version including feedback and brief rest. We assessed subjective threat assessed before the task, at the end of control condition, after each 10 min paired block of MIST, and twice upon completion of the task at 30 min and 60 min after starting the experimental task (see *Figure 4*). We used visual analogue scales to measure subjective threat. Salivary cortisol and α-amylase (*Engert et al., 2011*) samples were taken at the same time points as the visual analogue scales. Heart rate and blood pressure recordings were taken at 3 min intervals during the control MIST and two experimental MIST conditions, with four readings in each of these three conditions.

## Power calculation

In a study of test-retest reliability of [$^{18}$F]-DOPA PET (*Egerton et al., 2010*) striatal $K_i^{cer}$ had an intra-class correlation coefficient of approximately 0.9 [mean (SD) $K_i^{cer}$ = 0.01417 (0.00127)min$^{-1}$ (test) and 0.01381 (0.00127)min$^{-1}$ (re-test)]. Previous [$^{18}$F]-DOPA uptake work has found an effect size of 1.25 in patients with schizophrenia (Howes et al.) which compares well with previous studies: 1.89 (*Meyer-Lindenberg et al., 2002*), 1.57 (*McGowan et al., 2004*). On the basis that large effect sizes are observed in disorders of dopamine function, this study was powered to anticipate an effect size of $d$ = 1.00 when comparing differences between HA and LA groups. Therefore, to achieve a power of 0.8, with an effect size of 1.0, $a$ = 0.05, using independent $t$-tests, $n$ = 17 participants would be required per group.

## Statistical analysis

Normality of distribution and homogeneity of variance were assessed using Kolmogorov–Smirnov and Levene's tests respectively, and diagnostic plots. The primary analysis was for Group (HA/LA) differences in striatal dopamine synthesis capacity. The primary region of interest was the whole striatum. Exploratory analyses were performed in the functional subdivisions of the striatum. Independent samples $t$-tests were used for normally distributed data, Mann–Whitney $U$-tests for non-normally distributed data, and the $\chi^2$ test for dichotomous variables. Where the assumption of homogeneity of variance was violated for independent samples $t$-tests, p values were adjusted to assume unequal variance. Group differences in acute response to psychosocial stress were investigated by calculating the Area Under the Curve (AUC) using the trapezoid method. The AUC was calculated from the last measurement taken during the control MIST condition up until the last available post-task measurement. For subjective ratings (VAS), salivary cortisol and amylase, AUC was calculated from readings taken from control MIST (10 min), Active MIST 1 (20 min), Active MIST 2 (30 min), Post Active MIST 2 (40 min) and Post Task (70 min). For blood pressure (MAP), AUC was calculated from Control MIST (19 min), Active MIST 1 (22 minutes, 25 min, 28 min), Active MIST 2 (31 minutes, 34 min, 37 min), and Post Active MIST 2 (40 min). Prior to calculating the AUC, data were corrected for baseline performance by subtracting baseline scores from all subsequent time points. Independent sample $t$-tests were used to compare groups at baseline, to compare response to acute psychosocial stress using AUC. Pearson correlational analyses were conducted separately in each of the groups among variables showing group differences. Bivariate outliers (Cook's distance >1) were excluded prior to correlational analyses.

## Acknowledgements

We are very grateful to Yvonne Lewis and everyone at Imanova (now Invicro), Professor Vivette Glover, Drs David Bonsall and Dr Lucia Valmaggia for their assistance with this study. This work was funded by a Medical Research Grant to Professor Howes. Dr Bloomfield is funded by a UCL Excellence Fellowship and supported by the National Institute for Health Research University College London Hospitals Biomedical Research Centre.

# Additional information

## Competing interests

Oliver Howes: Professor Howes has received investigator-initiated research funding from and/or participated in advisory/speaker meetings organised by Astra-Zeneca, Autifony, BMS, Eli Lilly, Heptares, Jansenn, Lundbeck, Lyden-Delta, Otsuka, Servier, Sunovion, Rand and Roche. Neither Professor Howes nor his family have been employed by or have holdings/a financial stake in any biomedical company. The other authors declare that no competing interests exist.

## Funding

| Funder | Grant reference number | Author |
|---|---|---|
| Medical Research Council | MC-A656-5QD30 | Oliver Howes |
| National Institute for Health Research | | Michael AP Bloomfield |
| National Institute for Health Research | University College London Hospitals Biomedical Research Centre | Michael AP Bloomfield |
| Wellcome Trust | 094849/Z/10/Z | Oliver Howes |
| University College London | | Michael AP Bloomfield |
| National Institute for Health Research | Biomedical Research Centre at South London and Maudsley NHS Foundation Trust and King's College London | Oliver Howes |

The funders had no role in study design, data collection and interpretation, or the decision to submit the work for publication.

## Author contributions

Michael AP Bloomfield, Conceptualization, Data curation, Formal analysis, Investigation, Methodology, Writing—original draft, Project administration, Writing—review and editing; Robert A McCutcheon, Investigation, Writing—review and editing; Matthew Kempton, Methodology, Writing—review and editing; Tom P Freeman, Formal analysis, Writing—review and editing; Oliver Howes, Conceptualization, Resources, Formal analysis, Supervision, Funding acquisition, Investigation, Methodology, Writing—original draft, Writing—review and editing

## Author ORCIDs

Michael AP Bloomfield (iD) https://orcid.org/0000-0002-1972-4610
Robert A McCutcheon (iD) https://orcid.org/0000-0003-1102-2566

## Ethics

Human subjects: This study was approved by the National Research Ethics Service (12/LO/1955) and the Administration of Radioactive Substances Advisory Committee (ARSAC). The study was conducted in accordance with the Helsinki Declaration. All participants provided informed written consent to participate after an oral and written explanation of the study.

## Decision letter and Author response

Decision letter https://doi.org/10.7554/eLife.46797.013
Author response https://doi.org/10.7554/eLife.46797.014

## Additional files

### Supplementary files
• Source data 1. Processed data for *Figure 1*, *Figure 2* and *Table 2*.
DOI: https://doi.org/10.7554/eLife.46797.010
• Transparent reporting form DOI: https://doi.org/10.7554/eLife.46797.011

### Data availability
The raw data from this study are available on written request to the Chief Investigator. This restriction is due to sensitive data on human research participants. Processed data files for Figures 1 and 2, and table 2 are provided in Source data 1.

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
