## [Decision Letter]

Thank you for submitting your article "The effects of chronic psychosocial stress on dopaminergic function and psychiatric imaging group" for consideration by *eLife*. Your article has been reviewed by three peer reviewers, and the evaluation has been overseen by Christian Büchel as the Senior and Reviewing Editor. The reviewers have opted to remain anonymous.

The reviewers have discussed the reviews with one another and the Reviewing Editor has drafted this decision to help you prepare a revised submission.

Summary:

This paper is on chronic psychosocial adversity and vulnerability to mental illness. The authors follow the lead that dopamine might be an important link here and used [^18^ F]-DOPA PET to compare dopamine synthesis capacity of participants with high cumulative exposure to psychosocial adversity and controls (both groups n=17). The results show (i) that DA synthesis was correlated with subjective threat and physiological response to acute psychosocial stress, (ii) psychosocial stress dampened striatal dopaminergic and (iii) psychosocial adversity blunted physiological subjective responses to acute psychosocial stress.

Essential revisions:

Apart from addressing all points raised by the reviewers (see below), the authors should put particular emphasis on the following topics:

1) Explain and justify why the stress task (MIST) was performed two hours prior to the PET scanning.

2) How can acute control and experimental effects be disentangled and how is the interaction with chronic adversity resolved?

3) The sections regarding basic science and dopamine population activity needs clarifications.

Reviewer #1:

In this manuscript, the authors examine the impact of prior exposure to psychosocial stress on indices of DA function in humans. The paper is quite interesting and potentially important; however, there are some issues that need to be addressed:

1) The link with animal literature as written is confusing with regard to DA release and stress vs. synthesis capacity in humans. One potential caveat is that overall release in animals in response to stressors may not be the critical variable; since the authors are studying synthesis capacity (i.e., number of terminals that are active), a better comparison would be with DA neuron population activity (i.e., number of neurons firing) which was shown in studies of repeated stress by Valenti and Grace. These data would be more consistent and less confusing than the literature cited, which confounds release, metabolites, and activation.

2) Which regions of the DA system are activated vs. blunted with acute vs. chronic stress is an important variable. Acute and repeated stress activate the entire DA system projecting to much of the striatum (Valenti and Grace), in particular the associative striatum where object salience is important, whereas in chronic stress-induced depression (Holly and Miczek), the blunting occurs primarily in the neurons projecting to the ventromedial striatum (Moreines and Grace), where reward-related variables are processed. Therefore these are different systems mediating the DA stress response that varies with duration of stress exposure, and with the induction of anxiety (acute or repeated stress) vs. depression (chronic stress). This is consistent with Koob's opponent process model, where acute stress activates the DA system, which upon chronic exposure leads to a compensatory down-regulation.

3) Stress in Juvenile/adolescence has very different effects than chronic stress in adults. Therefore it is not valid to compare repeated adult stress with juvenile stress factors.

4) Discussion interpretation of findings – Holly and Miczek did not record from DA neurons.

5). The section on maternal deprivation occurs during very early juvenile period (i.e., postnatal day 7-8), which is far earlier than the juvenile/adolescent stress referred to in the paper.

Reviewer #2:

The authors present findings on dopamine synthesis capacity (using PET) in individuals with high exposure to adversity as compared to matched controls with low adversity (each group with N=17). PET scans are performed two hours after an acute psychosocial stress task (MIST), were participants with high exposure to adversity show enhanced threat experience, but lower arterial blood-pressure and trend-wise lower cortisol responses.

I am intrigued by these findings, since the author link a basal neurotransmitter function to cumulative adversity across different experiences in a non-clinical population. My excitement is however damped due to the order of the MIST and the PET scans: If the authors expect an effect of stressful events (chronically) on dopamine function, the PET scans would also be influenced (acutely) by the preceding stress-experience.

The authors need to clarify this effect. Since the design cannot be changed, the least would be a clear statement in the Abstract and main text.

The authors need to provide a power analyses. This study has a low number of subjects (I know that this might be common for PET studies), hence the authors need to show convincingly that the effects are well powered.

An additional factor, which needs to be considered (e.g., by covariation) is the amount of smoking in both groups. Even though the numbers of smokers is equal in both groups, the amount of smoking differs and has been shown to influence dopamine neurotransmission.

In the mediation analysis, the limbic subdivision of the striatum was chosen (subsection “The relationships between physiological and subjective measures”). The motivation for this subregion in the text ("connectivity to the threat system") and in the Materials and methods ("Exploratory analyses were performed in the functional subdivisions of the striatum.") is not very strong. I would suggest to perform regression analyses to identify which subregion is actually the best predictor for the physiological measures.

Reviewer #3:

In the study described by Bloomfield and colleagues, authors wanted to investigate whether exposure to chronic psychosocial stress, as assessed through a combination of childhood and adult stressors, would be associated with dopamine synthesis capacity. Authors exposed 34 subjects (17 with high and 17 with low cumulative stress exposure) to the Montreal Imaging stress task (MIST) and measured subjective, physiological and endocrine stress responses; two hours later, subjects were scanned for their dopamine synthesis capacity using 18F-Dopa in a Positron Emission Tomography experiment. Authors report that dopamine synthesis correlated with subjective threat and physiological response to acute psychosocial stress, and that high cumulative stress dampened striatal dopaminergic function.

Any PET study is a serious investment in time and resources and therefore authors should be commended to have undertaken this endeavor. The research question is not completely new as previous studies have investigated the possible role of stress in dopamine function, also using PET (and the authors make reference to this). What is unusual is the combination of early life and adult stressors to create a high cumulative lifetime adversity group; I have not come across this before. One can certainly argue whether this represents a useful approach as it confounds the early life (likely programming) effects with the later in life acute stressors which occur past the critical development periods. In fact, some authors (e.g., Ellis and colleagues with their work on life history theory) argue that early life adversity changes the stress systems to better prepare you for the adult life that is likely to be similar. In that sense, early life stress that is followed by adult stress might thus represent the expected outcome for the organism, that it is now better prepared for.

Be that as it may, I would simply have wished for a stronger justification for the particular chosen experimental approach as I was unclear on what exactly made the authors select these criteria. Also, why minority ethnic status was considered an adult stressor per se which was considered to be on par with bereavement or unemployment was not immediately clear to me and I would like to see more references justifying this approach.

What might be a potentially bigger concern had to do with the methodological approach chosen by the authors – if I understood the Materials and methods section correctly, authors did the stress task (the MIST) two hours prior to the PET scanning. Is the assumption that the stress would have the effect on the dopamine synthesis two hours after stress cessation? If that was the case, this should be made very explicit to the reader, and then I would like to see a section in the manuscript where this is systematically explained; right now, the reader is left in the dark about the choice of this exact rationale. Further, there was no stress – control comparison; subjects performed the first block of control followed by directly the experimental condition of the MIST; how can the authors be sure that what they then observe in the PET is specific to the stress part of the experimental manipulation? This is completely unclear to me, and I (and I am assuming the average reader as well) would benefit from a more thorough explanation of this part of the study.

[Editors' note: further revisions were requested prior to acceptance, as described below.]

Thank you for resubmitting your work entitled "The Effects of Chronic Psychosocial Stress on Dopaminergic Function and the Acute Stress Response" for further consideration at *eLife*. Your revised article has been favorably evaluated by Christian Büchel as the Senior and Reviewing Editor, and three reviewers.

The manuscript has been improved but there is one remaining issue that needs to be addressed before acceptance:

The data does not allow to say that DA synthesis is driven by early adversity alone, because it could be the interaction of childhood adversity and acute stress manipulation. The reviewers and the editor think that this does not lower the importance of this paper, but that it needs to be made clear in the paper, including the title (see below).

Reviewer #2:

The author responded to my comments and made an effort to clarify critical points in the manuscript.

The authors agree that the current study cannot disentangle chronic and acute stress, which is why I will consult with the other reviewers if the term "chronic" should be removed/changed from the title.

Reviewer #3:

The manuscript has benefited from the revision and now appears substantially improved. The authors have addressed all major points raised previously.

For the major points, the first point that was raised by this reviewer, i.e. the question of why a high cumulative lifetime adversity group was created, was answered thoroughly and convincingly, resulting in helpful revision in both the Introduction and the Discussion.

The second point previously raised, i.e. why a stress task preceded the PET scan by two hours, was answered by stating that the rationale was to reduce the variance in time. Here I am less enthusiastic that this is a sufficient response. Why would it be important to reduce the variation in time? On the other hand, by putting these two tests so close to each other, my fear would be that potential after effects from the stressor might have influenced what was observed during the PET scan. Authors do clarify that they are not aware of any evidence indicating that an acute stressor can have an effect on the parameters looked at in the PET, but then again, a controlled study comparing stress and no stress prior to PET to investigate the effects probably does not exist at this point, so obviously it is not known. Any number of things can occur two hours after an acute stressor – e.g., immune system changes, genomic effects of glucocorticoids, refractory period of hpa axis responsivity, which might or might not have an effect on the subsequent PET scan.

Thus, the decision to induce acute stress so close before the PET scan is in my opinion problematic, in the absence of a good rationale. Just to avoid the impression that it isn't, and to prevent other researchers from copying this design, it might be worth adding that having more time pass after an acute stressor might have been the superior choice.

---

## [Author Response]

Essential revisions:Apart from addressing all points raised by the reviewers (see below), the authors should put particular emphasis on the following topics:1) Explain and justify why the stress task (MIST) was performed two hours prior to the PET scanning.

Please see our response to reviewer 2 (below).

2) How can acute control and experimental effects be disentangled and how is the interaction with chronic adversity resolved?

We have addressed the reviewers comments relating to this below. We have commented on this in the Discussion:

“However, a potential limitation of our combination of early developmental and adult stressors is that if may confound the early life (i.e. likely programming) effects with the later in life acute stressors which occur after developmental sensitive periods. […] Future work is therefore needed to disentangle the effects of acute vs. chronic stressors on the dopamine system.”

3) The sections regarding basic science and dopamine population activity needs clarifications.

We have clarified these sections as requested by the reviewers.

Reviewer #1:In this manuscript, the authors examine the impact of prior exposure to psychosocial stress on indices of DA function in humans. The paper is quite interesting and potentially important; however, there are some issues that need to be addressed:1) The link with animal literature as written is confusing with regard to DA release and stress vs. synthesis capacity in humans. One potential caveat is that overall release in animals in response to stressors may not be the critical variable; since the authors are studying synthesis capacity (i.e., number of terminals that are active), a better comparison would be with DA neuron population activity (i.e., number of neurons firing) which was shown in studies of repeated stress by Valenti and Grace. These data would be more consistent and less confusing than the literature cited, which confounds release, metabolites, and activation.

Thank you for this suggestion. We have separated the animal from the human work. We have removed studies investigating stress-induced dopamine release from the animal work and added the following text to the paragraph on animal work:

“Animal research has demonstrated that acute stressors including aversive stimuli induce a pronounced activation of the dopamine system in terms of dopamine neuron population activity (i.e. the numbers of neurons firing) and with regards to amphetamine-induced behaviours (Valenti et at.).”

2) Which regions of the DA system are activated vs. blunted with acute vs. chronic stress is an important variable. Acute and repeated stress activate the entire DA system projecting to much of the striatum (Valenti and Grace), in particular the associative striatum where object salience is important, whereas in chronic stress-induced depression (Holly and Miczek), the blunting occurs primarily in the neurons projecting to the ventromedial striatum (Moreines and Grace), where reward-related variables are processed. Therefore these are different systems mediating the DA stress response that varies with duration of stress exposure, and with the induction of anxiety (acute or repeated stress) vs. depression (chronic stress). This is consistent with Koob's opponent process model, where acute stress activates the DA system, which upon chronic exposure leads to a compensatory down-regulation.

Thank you very much indeed for raising these important points. We have added the following text to the Discussion:

“There is evidence of regional specificity in the direction of effects of acute vs. chronic stress exposure. […] Our findings are consistent with Koob's opponent process model, where acute stress activates the dopamine system, which upon chronic exposure leads to a compensatory down-regulation (Koob et al., 1997)”.

3) Stress in Juvenile/adolescence has very different effects than chronic stress in adults. Therefore it is not valid to compare repeated adult stress with juvenile stress factors.

We have edited the relevant section of the Discussion to make it clearer when we are referring to developmental vs. adult stress exposure. We have revised the text as follows:

"Our results extend these findings and are consistent with evidence from anima/ models whereby subcortical dopamine transmission is blunted in response to multiple stressors in adulthood (Chrapusta, Wyatt and Masserano, 1997; Gresch et al., 1994)... Whilst stress exposure in animals during the juvenile period and adolescence has a very different effect from to chronic stress in adulthood, our findings are also broadly consistent with developmental stress models (Brake et al., 2004). "

4) Discussion interpretation of findings – Holly and Miczek did not record from DA neurons.

Reference to the review by Holly and Miczek has been replaced with reference to the study by Saal et al., 2003.

5). The section on maternal deprivation occurs during very early juvenile period (i.e., postnatal day 7-8), which is far earlier than the juvenile/adolescent stress referred to in the paper.

We have made the timing of maternal deprivation studies clear in the text:

"Likewise, early maternal deprivation models in the very early juvenile period (from post-natal day 5) have been associated with hypodopaminergic behaviours…”

Reviewer #2:The authors present findings on dopamine synthesis capacity (using PET) in individuals with high exposure to adversity as compared to matched controls with low adversity (each group with N=17). PET scans are performed two hours after an acute psychosocial stress task (MIST), were participants with high exposure to adversity show enhanced threat experience, but lower arterial blood-pressure and trend-wise lower cortisol responses.I am intrigued by these findings, since the author link a basal neurotransmitter function to cumulative adversity across different experiences in a non-clinical population.My excitement is however damped due to the order of the MIST and the PET scans: If the authors expect an effect of stressful events (chronically) on dopamine function, the PET scans would also be influenced (acutely) by the preceding stress-experience. The authors need to clarify this effect. Since the design cannot be changed, the least would be a clear statement in the Abstract and main text.

The study design was the same for both groups i.e. both groups underwent PET scans on the same day as acute psychosocial stress. Whilst we cannot exclude the possibility of an interaction (see below), it is unlikely that acute changes in dopamine synthesis are driving our results.

The Abstract has been amended as follows:

"The PET scan took place two hours after the induction of acute psychosocial stress using the Montréal Imaging Stress Task to induce acute psychosocial stress."

We have also added the following text in the limitations section of the Discussion:

"The PET scan was conducted following exposure to acute psychosocial stress exposure. Whilst this increases dopamine release (Pruessner et al., 2004), it is not clear if this has an acute effect on striatal dopamine synthesis capacity. […] A further study is needed to determine if acute stress alters dopamine synthesis capacity.”

The authors need to provide a power analyses. This study has a low number of subjects (I know that this might be common for PET studies), hence the authors need to show convincingly that the effects are well powered.

We have added a power calculation as follows:

"In a study of test-retest reliability of [^18^F]-DOPA PET (Egerton et al., 2010) striatal Kp^er^ had an intraclass correlation coefficient of approximately 0.9 [mean (SD) = 0.01417(0.00127)min-1 (test) and 0.01381(0.00127)min-1 (re-test)]. […] Therefore, to achieve a power of 0.8, with an effect size of 1.0, a=0.05, using independent t-tests, n=17 participants would be required per group. "

An additional factor, which needs to be considered (e.g., by covariation) is the amount of smoking in both groups. Even though the numbers of smokers is equal in both groups, the amount of smoking differs and has been shown to influence dopamine neurotransmission.

The following has been inserted into the text:

"As the amount of smoking differed in the groups and heavy smoking can influence dopamine function (Bloomfield et al., 2014; Salokangas et al., 2000) we performed an ANCOVA to examine whether smoking was influencing our findings. When co-varying for amount of current cigarette use, the group difference remained significant in the limbic striatum only 5.2, p=.029, "Ép=. 15)."

In the mediation analysis, the limbic subdivision of the striatum was chosen (subsection “The relationships between physiological and subjective measures”). The motivation for this subregion in the text ("connectivity to the threat system") and in the Materials and methods ("Exploratory analyses were performed in the functional subdivisions of the striatum.") is not very strong. I would suggest to perform regression analyses to identify which subregion is actually the best predictor for the physiological measures.

The limbic striatum was chosen because of its role in the stress response, its functional connectivity to the threat detection system and the largest effects sizes for group differences observed in our study. However, regression analyses did not identify which sub-region was the best predictor of the physiological measures (p>.08). We have therefore removed the mediation analysis from the manuscript and inserted the following into the text:

"Regression analyses did not identify which striata/ sub-region was the best predictor of the physiological measures (p>.08)".

Reviewer #3:[…] Any PET study is a serious investment in time and resources and therefore authors should be commended to have undertaken this endeavor. The research question is not completely new as previous studies have investigated the possible role of stress in dopamine function, also using PET (and the authors make reference to this).What is unusual is the combination of early life and adult stressors to create a high cumulative lifetime adversity group; I have not come across this before. One can certainly argue whether this represents a useful approach as it confounds the early life (likely programming) effects with the later in life acute stressors which occur past the critical development periods. In fact, some authors (e.g., Ellis and colleagues with their work on life history theory) argue that early life adversity changes the stress systems to better prepare you for the adult life that is likely to be similar. In that sense, early life stress that is followed by adult stress might thus represent the expected outcome for the organism, that it is now better prepared for.Be that as it may, I would simply have wished for a stronger justification for the particular chosen experimental approach as I was unclear on what exactly made the authors select these criteria. Also, why minority ethnic status was considered an adult stressor per se which was considered to be on par with bereavement or unemployment was not immediately clear to me and I would like to see more references justifying this approach.

We are grateful to the reviewer for raising these important points. Findings from epidemiological studies have found that early developmental adversity increases the risk of psychopathology in response to stressors in adulthood (e.g. McLaughlin et al., 2010) are not consistent with the theory that experiencing childhood trauma is protective against risk of mental illness in response to adult stressors.

We have revised the justification for our experimental approach, including the rationale for including minority ethnic status, in the Introduction as follows:

"Studies of psychosocial stressors and dopamine function have typically investigated risk factors in isolation, despite the fact that the risk factors cluster together and may share common underlying mechanisms (Hjern, Wicks, and Dalman; Morgan and Fisher; Wicks et al., 2005). […] Given the findings of dopaminergic dysfunction associated with childhood maltreatment presented above, we hypothesised that healthy humans with a high cumulative exposure to psychosocial stressors would have altered striatal dopamine synthesis, compared to humans with a low exposure."

We have added the following to the Discussion:

"We chose to recruit participants with high levels of stress exposure in early development and adulthood, because early developmental stressors increase the risk of psychopathology following adult stressors (McLaughlin et al., 2010). However, a potential limitation of our combination of early developmental and adult stressors is that it may confound the early life (i.e. likely programming) effects with the later in life acute stressors which occur after developmental sensitive periods. "

What might be a potentially bigger concern had to do with the methodological approach chosen by the authors – if I understood the Materials and methods section correctly, authors did the stress task (the MIST) two hours prior to the PET scanning. Is the assumption that the stress would have the effect on the dopamine synthesis two hours after stress cessation? If that was the case, this should be made very explicit to the reader, and then I would like to see a section in the manuscript where this is systematically explained; right now, the reader is left in the dark about the choice of this exact rationale. Further, there was no stress – control comparison; subjects performed the first block of control followed by directly the experimental condition of the MIST; how can the authors be sure that what they then observe in the PET is specific to the stress part of the experimental manipulation? This is completely unclear to me, and I (and I am assuming the average reader as well) would benefit from a more thorough explanation of this part of the study.

We apologise that this was not clear enough. The rationale for conducting the PET scan on the same day as the stress task was to reduce the variance in the time between the measures. We did not have assumptions that acute stress exposure would alter subsequent dopamine synthesis capacity. Whilst we cannot exclude the possibility that such effects may be contributing to our results, we are not aware of evidence indicating that an acute stressor can alter two parameters that contribute to our index (i.e. the number of available dopamine vesicles and/or the activity of aromatic acid decarboxylase) in this timeframe. The rationale for exploring the relationships between stress-induced measures and dopamine synthesis capacity is implicit is provided in the manuscript. We have included these points as follows:

Materials and methods:

"The rationale for conducting the PET scan on the same day as the stress task was to reduce the variance in the time between the measures."

Discussion:

"However, we are not aware of evidence indicating that an acute stressor can alter the parameters that contribute to our index (i.e. the activity of aromatic acid decarboxylase) in the timeframe used in our study. "

[Editors' note: further revisions were requested prior to acceptance, as described below.]The manuscript has been improved but there is one remaining issue that needs to be addressed before acceptance:The data does not allow to say that DA synthesis is driven by early adversity alone, because it could be the interaction of childhood adversity and acute stress manipulation. The reviewers and the editor think that this does not lower the importance of this paper, but that it needs to be made clear in the paper, including the title (see below).

We are grateful to Professor Buchel and colleagues for their further review. We have made changes to the paper including the title.

Reviewer #2:The author responded to my comments and made an effort to clarify critical points in the manuscript.The authors agree that the current study cannot disentangle chronic and acute stress, which is why I will consult with the other reviewers if the term "chronic" should be removed/changed from the title.

We are grateful to the reviewer. We have removed the term "chronic" from the title. The title now reads: "The Effects of Psychosocial Stress on Dopaminergic Function and the Acute Stress Response".

Reviewer #3:[…] For the major points, the first point that was raised by this reviewer, i.e. the question of why a high cumulative lifetime adversity group was created, was answered thoroughly and convincingly, resulting in helpful revision in both the Introduction and the Discussion.The second point previously raised, i.e. why a stress task preceded the PET scan by two hours, was answered by stating that the rationale was to reduce the variance in time. Here I am less enthusiastic that this is a sufficient response. Why would it be important to reduce the variation in time? On the other hand, by putting these two tests so close to each other, my fear would be that potential after effects from the stressor might have influenced what was observed during the PET scan. Authors do clarify that they are not aware of any evidence indicating that an acute stressor can have an effect on the parameters looked at in the PET, but then again, a controlled study comparing stress and no stress prior to PET to investigate the effects probably does not exist at this point, so obviously it is not known. Any number of things can occur two hours after an acute stressor – e.g., immune system changes, genomic effects of glucocorticoids, refractory period of hpa axis responsivity, which might or might not have an effect on the subsequent PET scan.Thus, the decision to induce acute stress so close before the PET scan is in my opinion problematic, in the absence of a good rationale. Just to avoid the impression that it isn't, and to prevent other researchers from copying this design, it might be worth adding that having more time pass after an acute stressor might have been the superior choice.

We are grateful to the reviewer for their feedback. In performing the PET scan so close to the stress task we sought to reduce the length of time between task and PET scan to reduce variance associated with normal fluctuations in dopamine synthesis capacity. We agree is possible that close temporal proximity of the stress task to the PET means that it is possible that what was observed during the PET scan could have been influenced by the stress task. Whilst we not aware of any evidence indicating that an acute stressor can have an effect on the parameters looked at in the PET, a controlled study comparing stress and no stress prior to PET to investigate the effects is needed to definitively address this possibility. Given this, it is possible therefore that an optimal experimental design would allow more time to pass between the stressor and the PET scan. These points have been clarified in the text as below:

"The PET scan was conducted following exposure to acute psychosocial stress exposure. […] Nonetheless, a controlled study comparing stress and no stress prior to PET to investigate the effects is needed to definitively determine if acute stress alters dopamine synthesis capacity. Given this, it is possible that an optimal experimental design would allow more time to pass between the stressor and the PET scan.”